# Pseudogene *TNXA* Variants May Interfere with the Genetic Testing of CAH-X

**DOI:** 10.3390/genes14020265

**Published:** 2023-01-19

**Authors:** Qizong Lao, Kiet Zhou, Megan Parker, Fabio R. Faucz, Deborah P. Merke

**Affiliations:** 1National Institutes of Health Clinical Center, Bethesda, MD 20892, USA; 2The Eunice Kennedy Shriver National Institute of Child Health and Human Development, National Institutes of Health, Bethesda, MD 20892, USA

**Keywords:** congenital adrenal hyperplasia, Ehlers–Danlos syndrome, CAH-X, EDS, *TNXB*

## Abstract

CAH-X is a hypermobility-type Ehlers–Danlos syndrome connective tissue dysplasia affecting approximately 15% of patients with 21-hydroxylase deficiency (21-OHD) congenital adrenal hyperplasia (CAH) due to contiguous deletion of *CYP21A2* and *TNXB* genes. The two most common genetic causes of CAH-X are *CYP21A1P-TNXA/TNXB* chimeras with pseudogene *TNXA* substitution for *TNXB* exons 35–44 (CAH-X CH-1) and *TNXB* exons 40–44 (CAH-X CH-2). A total of 45 subjects (40 families) from a cohort of 278 subjects (135 families of 21-OHD and 11 families of other conditions) were found to have excessive *TNXB* exon 40 copy number as measured by digital PCR. Here, we report that 42 subjects (37 families) had at least one copy of a *TNXA* variant allele carrying a *TNXB* exon 40 sequence, whose overall allele frequency was 10.3% (48/467). Most of the *TNXA* variant alleles were in *cis* with either a normal (22/48) or an In2G (12/48) *CYP21A2* allele. There is potential interference with CAH-X molecular genetic testing based on copy number assessment, such as with digital PCR and multiplex ligation-dependent probe amplification, since this *TNXA* variant allele might mask a real copy number loss in *TNXB* exon 40. This interference most likely happens amongst genotypes of CAH-X CH-2 with an in *trans* normal or In2G *CYP21A2* allele.

## 1. Introduction

Congenital adrenal hyperplasia (CAH) due to 21-hydroxylase deficiency (21-OHD; OMIM 201910) is an autosomal recessive disease of steroidogenesis that affects 1 in 15,000 in its severe classic form and 1 in 200 in its mild nonclassic form, with asymptomatic carrier prevalence estimated as 1 in 60 and 1 in 11, respectively [1,2,3]. Manifestations consistent with hypermobility-type Ehlers–Danlos syndrome connective tissue dysplasia (hEDS; OMIM 130020) are often found amongst CAH patients, with tenascin-X defects as the most commonly reported etiology [4,5,6,7,8,9]. Recent studies of large CAH cohorts in different continents revealed that approximately 15% of the CAH population has hEDS due to a contiguous gene deletion affecting both the *CYP21A2* and *TNXB* genes [10,11,12,13], which warrants re-examination of current CAH clinical and genetic evaluation workflows [14,15,16].

The genes *CYP21A2,* encoding 21-hydroxylase, and *TNXB,* encoding tenascin-X, are mapped to the major histocompatibility complex on chromosome 6 (p21.33), with their respective 3′-UTR and last exon (exon 44) overlapping. Their locus is a typical low-copy repeat and is termed *RCCX* module, standing for “*RP-C4-CYP21-TNX*” in tandem. *RP* signifies *RP1* encoding a serine/threonine nuclear protein kinase and pseudogene *RP2* (synonyms for *STK19* and *STK19P*, respectively); *C4* signifies *C4A* and *C4B* encoding two complement component 4 isotopes; *CYP21* signifies *CYP21A2* and pseudogene *CYP21A1P*, while *TNX* signifies *TNXB* and pseudogene *TNXA*. These gene pairs are highly homologous, making the *RCCX* locus vulnerable to gene conversions and unequal crossovers, often resulting in pathogenic variants [17]. In CAH, the vast majority of pathogenic alleles are due to two mechanisms: pseudogene minor conversion, resulting in 12 common variants of *CYP21A1P* origin and unequal crossovers, resulting in chimeric genes (also termed 30-kb deletions), accounting for 60% and 30% of allele frequency, respectively [3,18,19,20,21]. The chimeric genes can be further categorized into CAH (*CYP21A1P/CYP21A2*) and CAH-X (*CYP21A1P-TNXA/TNXB*) chimeras (Figure 1) [22,23], affecting *CYP21A2* only or both *CYP21A2* and *TNXB*, respectively. *CYP21A1P-TNXA/TNXB* chimeras are responsible for the majority of known hEDS cases amongst the CAH population, known as “CAH-X”, named for CAH with tenascin-X defects [4,5,9].

EDS is a group of connective tissue disorders involving multiple genes related to collagen pathways and extracellular matrix (ECM) maintenance [24]. hEDS is the most common subtype, with an estimated prevalence of 1 in 5000 or higher, and the genetic basis is mostly unknown [25]. Concomitant hEDS in CAH has been associated with *TNXB*, which encodes tenascin-X, a glycoprotein crucial in ECM composition [5,6,26]. Variants/defects throughout the entire 68 kb *TNXB* gene have been reported to cause hEDS [27]; however, those related to CAH-X mostly enrich within a 4.1 kb span of the *TNXB* 3′-end (exons 32–44) that shares 92% homology with *TNXA* [7,8,10,11,12,13]. Two CAH-X chimeras account for the vast majority of CAH-X genotypes and also represent 30% of 30-kb deletions, or 15% of all CAH alleles: CAH-X CH-1 substitutes *TNXA* for *TNXB* exons 35–44, resulting in a null allele; while CAH-X CH-2 substitutes *TNXA* for *TNXB* exons 40–44, resulting in a dominant negative effect [8]. The *TNXA* substitution for *TNXB* at exon 35 features a 120 bp deletion (c.11435–11524+30del NM_019105.8) spanning exon and intron 35, whereas the *TNXA* substitution for *TNXB* at exon 40 features two contiguous variants of c.12150C>G (synonymous) and c.12174C>G (p.C4058W) (Figure 1). Unlike autosomal recessive CAH, the hEDS manifestation caused by these two CAH-X chimeras appears to be autosomal dominant regardless of the CAH phenotype. However, in general, CAH patients suffer more connective tissue dysplasia phenotype than their carrier counterparts, possibly due to the influence of chronic glucocorticoid exposure [7,8,11,28]. Other less common CAH-X defects include splice site variants in intron 42 and CAH-X CH3 with *TNXA* substitute for *TNXB* exons 41–44, [29,30].

The molecular genetic testing of *TNXB* is challenging. Although it is suitable for next-generation sequencing (NGS) platforms due to its 68 kb mass, the 3′-end (exons 32–44) has to rely on other methods due to the significant mapping interference by *TNXA* and its copy number variations [14,31]. Sanger is the most comprehensive and accurate methodology used for CAH and CAH-X genetic testing, but it is laborious and low-throughput [32]. We previously developed a high-throughput assay for fast and cost-effective screening of CAH-X chimeras (CH-1 and CH-2) so that laboratories can preserve Sanger for confirmation purposes [10]. The assay was an allele-specific PCR-based copy number assessment of *TNXB* exons 35 and 40: copy number losses in both exons warranted a call of CAH-X CH-1, whereas that in exon 40 alone warranted a call of CAH-X CH-2. The assay had an overall 99.3% (276/278) accuracy in our CAH cohort. Notably, we observed a total of 45 out of 234 true CAH-X negatives with a ≥2.5 measured copy number in *TNXB* exon 40, in contrast to only one true negative with an extra copy number measured in exon 35 (Figure 2). 

In this study, we sought to characterize the extra *TNXB* exon 40 copy number measured in true CAH-X negatives. We also investigated associations between this *TNXA* variant and various CAH alleles, and considered potential interference with current CAH-X genotyping methodologies to further understand the complexity across *RCCX* modules. 

## 2. Materials and Methods

This study was approved by the National Institutes of Health Institutional Review Board. All subjects were enrolled in an ongoing Natural History Study at the National Institutes of Health Clinical Center in Bethesda, MD, USA (NCT00250159). The cohort was composed of a total of 278 subjects (145 affected patients of 21-OHD CAH, 116 carriers of 21-OHD CAH, 3 normal relatives from 135 families, and 11 affected patients and 3 relatives of other diseases such as EDS, XY disorder of sex development and other CAH types from 11 families). All adult subjects or parents of children (<18 years old) gave written informed consent. Clinical evaluations for the diagnosis of CAH and EDS were completed as described previously [7]. All subjects had comprehensive genetic records, including a complete CAH and CAH-X genotype by a CLIA-accredited laboratory (PreventionGenetics, LLC, Marshfield, WI), as well as copy numbers of *TNXB* exons 35, 40, *C4A*, *C4B* genes determined as previously described [10,32,33]. For each subject, *TNXA* copy number was determined as “*C_TNXA_* = *C_C4A_* + *C_C4B_* − 2”, with 2 accounting for *TNXB* and/or CAH-X chimeras if applicable. For the analysis of *TNXA* locus homologous to *TNXB* exon 40, allele-specific PCR was conducted with the primer pair of 5′-GACCCAGAAACTCCAGGTGGGAG-3′ and 5′-CACCGAGAACTCGAAGCCCTTC-3′, followed by Sanger sequencing with primer 5′-CAATGAGGCCCTGCACAGC-3′. NC_000006.12 (Chr6: 32,008,420–32,013,023, complement) and NC_000006.12 (chr6: 32,041,153–32,109,338, complement) were used as reference for *TNXA* and *TNXB*, respectively. DNASTAR Lasergene (Madison, WI, USA) was used for sequence alignment analysis.

## 3. Results

Amongst the 278 subjects who underwent CAH-X screening, 45 of them from 37 unrelated families (31 with 21-OHD and 6 others) had an excessive copy number of *TNXB* exon 40 (2.50–5.47, with 2.50 as the cutoff value) as measured by the digital droplet PCR assay. These 45 subjects were all confirmed as true CAH-X negative with a Sanger-based methodology conducted by a CLIA-accredited laboratory (PreventionGenetics, LLC, Marshfield, WI, USA). Their disease conditions included 18 patients with 21-OHD CAH (9 salt-wasting, 8 simple virilizing and 1 nonclassic), 18 carriers of 21-OHD CAH, 3 unaffected relatives of 21-OH CAH and 6 relatives of subjects with other endocrine conditions (Table 1). Forty-two subjects (93.3%, 42/45) had at least one allele of a *TNXA* variant homologous to the *TNXB* exon 40 by carrying SNPs rs4959086(C>G) and rs77471377(C>G), forty-one of them with both SNPs in tandem and one subject with rs77471377(C>G) alone. A total of 48 *TNXB*-exon 40-like *TNXA* variant alleles were detected in these 42 subjects. All 278 subjects of the cohort had previously measured *C4A* and *C4B* copy numbers by which their respective *RCCX* module unit numbers were determined [33]; accordingly, their respective *TNXA* copy numbers were also determined. There were a total of 467 *TNXA* alleles in the entire cohort; therefore, the allele frequency of the *TNXA* variant allele responsible for the excessive ddPCR *TNXB* exon 40 copy number was 10.3% (48/467). Amongst the 11 subjects carrying one copy of *TNXA*, 10 had the *TNXB*-like variant as their lone *TNXA* allele. Amongst the 27 subjects carrying two copies of *TNXA*, 3 had both copies and 23 had one copy as the *TNXB*-like variant. Amongst the seven subjects carrying three copies of *TNXA*, one had all three copies, one had two copies and four had one copy as the *TNXB*-like *TNXA* variant. The two SNPs were determined as in *cis* in all heterozygous cases by either family genotyping or TA clone sequencing, reflecting the ddPCR methodology which detects SNPs in tandem. As a control group, twenty-nine subjects confirmed negative for CAH-X and a normal range *TNXB* exon 40 copy number (1.7–2.4) were tested on the *TNXA* locus of interest. No SNP rs77471377(C>G) was detected in this group, whereas four carried a heterozygous rs4959086(C>G).

Hence, this *TNXB*-like *TNXA* variant allele was indeed the predominant cause of the excessive *TNXB* exon 40 copy number measured by ddPCR. It was found most frequently in *cis* with a normal *CYP21A2* allele, with 22 out of 48 alleles, or 45.8% in *cis* with 21 (or 20) normal *CYP21A2* alleles in 19 subjects from 17 families. The second most frequent in *cis CYP21A2* genotype was In2G; 12 (25.0%, 12/48) of the *TNXA* variant alleles shared a chromosome with 11 *CYP21A2* In2G alleles in 10 subjects from six families. Other findings included: two families had a *TNXA* variant allele in *cis* with a *CYP21A2* Q318X allele; one family each had a *TNXA* variant in *cis* with a *CYP21A2* I172N, del-I172N and IV8 + 1G>A allele, respectively. We failed to determine the in *cis CYP21A2* association of seven *TNXA* variant alleles due to absence of family genotype information. Thus, based on the 150 chromosomes with known normal *CYP21A2* and the 93 chromosomes of In2G *CYP21A2* (55 patients with 21-OHD CAH and 24 carriers) in our cohort, we found that at least 13.3% (20/150) and 11.8% (11/93) of unaffected and In2G *CYP21A2* alleles shared a chromosome with one or more copies of this *TNXA* variant allele, respectively. 

Three subjects with excessive ddPCR *TNXB* exon 40 copy number did not have a detectable *TNXB*-like *TNXA* allele, as shown in Table 1. Subject 25A had her *TNXB* exon 40 copy number measured as 2.62; she had one copy of *TNXA* determined as wild-type which matched the reference genome in this study, thus the mechanism underlying the excessive 0.62 copy of *TNXB* exon 40 remains unclear. Subject 33 had the *TNXB* exon 40 copy number measured as 2.87; two wild-type copies of *TNXA* were present. This was the lone case of *TNXB* exon 35 being measured as 2.83 (Figure 2) without further evidence of structural variation in the loci; thus, the excessive copy numbers were likely due to some variants in the *HBB* gene whose copy number was used to normalize the *TNXB* exons in the ddPCR assays. Subject 27B had 2.92 copies of *TNXB* exon 40 measured by ddPCR with 3 copies of *TNXA,* but our PCR protocol repeatedly failed to amplify these loci, leaving the mechanism unknown.

## 4. Discussion

Large cohort studies of patients with CAH due to 21-OHD worldwide have revealed and confirmed that approximately 15% of CAH patients carry at least one CAH-X allele, a contiguous deletion disrupting both *CYP21A2* and *TNXB* tandem genes [10,11,12,13]. Moreover, these deletions account for approximate 30% of chimeric genes, also termed 30-kb deletions. While CAH due to *CYP21A2* defects is autosomal recessive, the hypermobile EDS connective tissue dysplasia due to *TNXB* defects appears to be autosomal dominant, as carrying one CAH-X allele has been associated with an EDS phenotype [7,8,10]. Other than typical joint and skin conditions, CAH-X patients have about 25% prevalence of cardiac abnormalities, including congenital heart defects, such as structural valve abnormalities, left ventricular diverticulum and patent foramen ovale [7,8]. A diagnosis of CAH-X is beneficial by offering awareness and early medical intervention if indicated, but traditional clinical diagnosis of CAH-X relying on joint hypermobility and subluxations is often restricted by factors such as age, and a reliable tenascin-X serum level assay is not available [34]. As an accurate and cost-effective option, we recently suggested the inclusion of CAH-X genotyping with the standard scope of CAH genetic tests, especially for individuals carrying a “30-kb deletion” genotype, given that a large portion (30%) of this genotype are in fact CAH-X chimeras [10]. We now describe potential interference with CAH-X molecular genetic testing based on copy number assessment due to a commonly found *TNXA* variant allele carrying a *TNXB* exon 40 sequence.

The test and evaluation of CAH-X should also extend to CAH carriers who lack medical attention regarding CAH-related conditions [14,15,16]. Although hEDS manifestations are often milder compared to the CAH-X patient, carriers of a CAH-X allele are at risk of developing significant conditions, including a cardiac structural defect. In fact, since the classic CAH carrier prevalence is 1 in 60, about one third of CAH alleles are 30-kb deletions and one third of these are estimated to be CAH-X chimeras; CAH-X amongst CAH carriers alone is estimated to affect 1 in 670 in the general population. This coincides with a previous presumption that the 1 in 5000 prevalence of hEDS was grossly underestimated [25].

In order to include CAH-X chimeras into an existing CAH genetic test, Sanger sequencing followed by an allele-specific PCR with CYP779F/Tena32F primer pair remains the most accurate and comprehensive methodology. It is suitable for all conditions across the *CYP21A2*-*TNXB* exons 32–44 locus, and the test of CAH-X chimeras can be a simple add-on to the existing scope. However, this methodology is low-throughput and often restricted by a difficult 8.5 kb long-range PCR with frequent heterogeneous insert–deletions across the locus [32]. Clinical laboratories also use high-throughput methodologies, such as quantitative PCR, multiplex mini-sequencing, conversion-specific PCR (MMCP) and multiplex ligation-dependent probe amplification (MLPA), to conduct CAH genetic testing [14,35,36]. Among these methodologies, only the MLPA platform tests the *TNXB* exon 35 copy number that can be used for calling CAH-X CH-1, but *TNXB* exon 40 is completely out of scope. In order to evaluate *TNXB* status, our recently developed ddPCR-based assay may be added to these CAH methodologies. Alternatively, new assays amenable to existing platforms could be developed, but in either case, the potential influence of the pseudogene *TNXA* homologue should be considered.

In our previous work of developing a high-throughput CAH-X screening assay based on *TNXB* copy number losses, we observed that a large portion of true CAH-X negative samples (45/234) had an excessive *TNXB* exon 40 copy number (≥2.5) as measured by ddPCR [10]. We previously hypothesized that this was likely caused by a *TNXA* variant allele carrying SNPs rs77471377(C>G) and rs4959086(C>G) in tandem, which converted the locus to be *TNXB*-like, and thus detectable by the ddPCR assay. This *TNXA* variant allele was, in fact, commonly found in our CAH cohort. Although it remains as a pseudogene without affecting phenotypic outcomes, this *TNXB*-like *TNXA* variant allele may interfere genetic tests of CAH-X based on *TNXB* copy number assessment. For example, when a subject carries a CAH-X CH-2 chimera with at least one copy of this *TNXA* variant, the real *TNXB* exon 40 copy number loss will be masked by the latter’s homologue, creating an inevitable risk of a false negative call on CAH-X CH-2. In our large CAH cohort, 31 out of 135 21-OHD CAH families and 6 out of 11 families with other conditions carried such a *TNXB*-like *TNXA* allele, suggesting this threat of false negative CAH-X CH-2 determinations might commonly exist. However, we did not record any false negative cases in the assessment of all 278 subjects in our cohort. One explanation could be that no such *TNXA* variant was found in *cis* with any of the CAH 30-kb deletion genotypes. Since CAH-X alleles are mostly *TNXA*-free, with mono-modular *RCCX* structure, this might significantly reduce the possibility of *TNXA*-sourced interferences. On the other hand, masking interference from the in *trans* chromosome warrants consideration. The majority of *TNXB*-like *TNXA* variants were in *cis* with either a normal or an In2G *CYP21A2* allele. Accordingly, CAH carriers with a CAH-X CH-2/normal genotype and CAH patients with a CAH-X CH-2/In2G genotype are more likely than other genotypes to encounter a false negative call. Theoretically, this chance is approximately 10%, given that the prevalence of this *TNXA* variant allele amongst chromosomes of normal and In2G *CYP21A2* was 13.3% and 11.8%, respectively. 

Although we did not have a true false negative call in our cohort, we did have an example of masking interference from the in *trans* chromosome (Figure 2, the outlier red dot). One CAH carrier with CAH-X CH-1/normal genotype had her *TNXB* exons 35 and 40 copy numbers measured as one and two, respectively, which is different from a typical one copy each of the two *TNXB* exons in a monoallelic CAH-X CH-1. We found that a *TNXB*-like *TNXA* allele in her normal chromosome was the cause. This would have been a case of false negative call if she had carried a CAH-X CH-2 allele instead of CAH-X CH-1, whose loss in *TNXB* exon 35 copy number alone warranted a correct call [10].

In the genetics of 21-OHD CAH, a minor conversion is defined as a small portion of the pseudogene *CYP21A1P* being converted to *CYP21A2* to cause pathogenic defects, which include the 10 most common variants [3,14]. These variants have been well investigated and are currently within the scope of all standard CAH test platforms [14]. On the other hand, gene conversion in the opposite direction, from *CYP21A2* to *CYP21A1P*, has only been documented in a few studies. Cantürk, C. et al. reported that 8.5% of *CYP21A1P* alleles are *CYP21A2*-like at the position corresponding to p.I172, which might be mistaken for a *CYP21A2* duplication when using MLPA methodology [37]. Tsai, L. et al. reported that *CYP21A2*-like spots were commonly found throughout *CYP21A1P*, such as p.P30, p.G110, p.I172, p.V281, p.Q318 and p.R356 [38,39,40]. Notably, these reports were all based on healthy non-CAH populations. Although these prior reports suggested potential pseudogene interference with *CYP21A2* genotyping, the true frequency or significance of this interference has not yet been established due to the lack of studies using large CAH cohorts. In our current study, we reveal that real-gene-like spots also exist in pseudogene *TNXA*, or in at least a locus corresponding to *TNXB* exon 40. This *TNXB*-like *TNXA* variant might cause a false negative call on CAH-X CH-2 if using high-throughput methods such as ddPCR. Our findings provide noteworthy insight into the nature and complexity of the *RCCX* modules.

Finally, our findings support the concept that standard NGS-based methodologies are not yet suitable for the genetic testing of *RCCX* module genes within the homologous window, not only for *CYP21A2* but also for *TNXB* exons 32–44 [14,31,41]. Mainstream population-based databases using NGS methodologies, such as the gnomAD browser, lists allele frequency of the two SNPs of this study to be 0.1795 for rs4959086(C>G) and 0.02 for rs77471377(C>G) [42]. However, since the copy number of *TNXA* varies from 0–4 in each chromosome, the classical calculation of allele frequency based on two alleles in each subject is not accurate. In fact, there were a total of 467 *TNXA* alleles in our cohort of 278 subjects; thus, an allele frequency of the *TNXA* variant haplotype with both SNPs in tandem was 10.3% (48/467), suggesting an underestimation of rs7741377(C>G) allele frequency in the CAH population. One may argue that findings could be different in the general population; we therefore also evaluated the frequency of *TNXA* variant haplotype exclusively amongst the “normal” non-CAH chromosomes in our cohort. Because the *TNXA* copy number was determined in diploid, while haploid-specific copy number was not available in all subjects, our next frequency calculation was based on “chromosome” rather than “allele”. There were a total of 150 normal chromosomes in our cohort; at least 20 of them carried one or more copies of the *TNXA* variant haplotype, and therefore the chromosome-based frequency should be approximately 13.3% (20/150), which again agreed with our prediction of an underestimated rs7741377(C>G) allele frequency. In either case, our results suggested that the real-gene-like variants throughout pseudogenes *CYP21A1P* and *TNXA*, as well as their underestimated allele frequency, should be considered as another obstacle to overcome in addition to the commonly accepted pseudogene homologue and copy number variations in terms of designing a comprehensive CAH test platform based on NGS methodologies.

## Figures and Tables

**Figure 1 genes-14-00265-f001:**
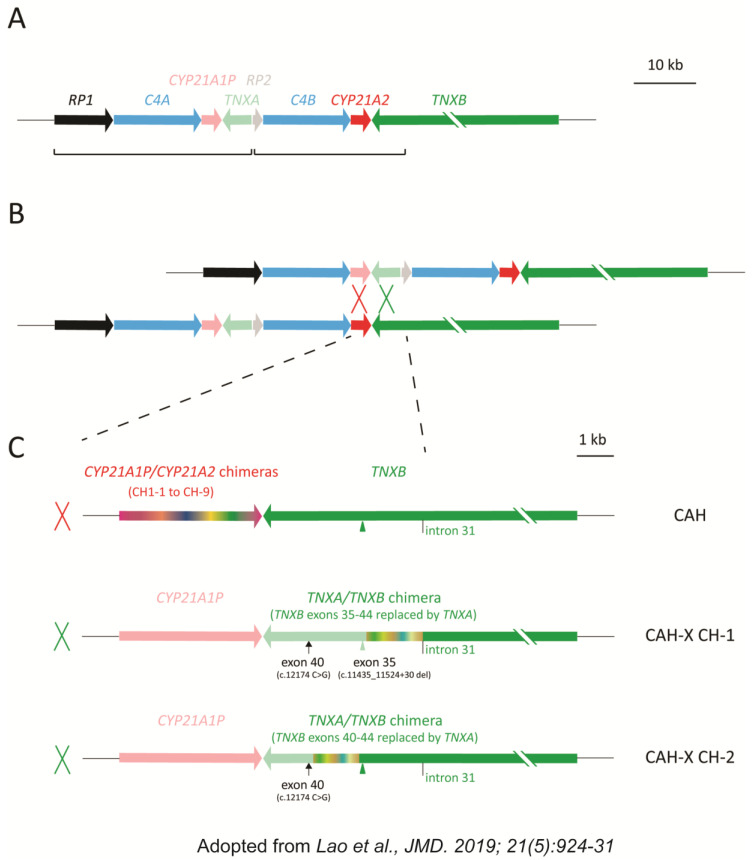
Schematic diagram of CAH and CAH-X chimeric genes [10]. (**A**), typical bimodular *RCCX* module: each pair of the homologous genes is shown in similar colors, with lighter colors representing pseudogenes; two *RCCX* units vulnerable for gene conversion are in brackets. (**B**), unequal crossover between *CYP21A1P* and *CYP21A2*, or *TNXA* and *TNXB*, results in a commonly termed “30-kb deletion” CAH genotype. (**C**), schemes of three major subtypes of “30-kb deletion”: top, *CYP21A1P/CYP21A2* chimeric genes with intact *TNXB*, pathogenic for CAH; middle, *CYP21A1P-TNXA/TNXB* chimera CAH-X CH-1 with *TNXB* exons 35–44 replaced by *TNXA* causes CAH-X due to tenascin-X haploinsufficiency; bottom, *CYP21A1P-TNXA/TNXB* chimera CAH-X CH-2 with *TNXB* exons 40–44 replaced by *TNXA* causes CAH-X due to a dominant negative effect. CAH-X CH-1 has an exon 35 c.11435_11524+30 deletion (light green triangle) and an exon 40 c.12174C>G mutation (small black arrow) in tandem, whereas CAH-X CH-2 has an intact exon 35 (green triangle) and an exon 40 c.12174C>G mutation. The junction site window for each chimeric gene is shown in chameleonic colors. Schemes from *CYP21A1P* to *TNXB* intron 31, which is the boundary of *RCCX* module homologous repeats, are shown in scale. The size of *TNXB* is 68 kb.

**Figure 2 genes-14-00265-f002:**
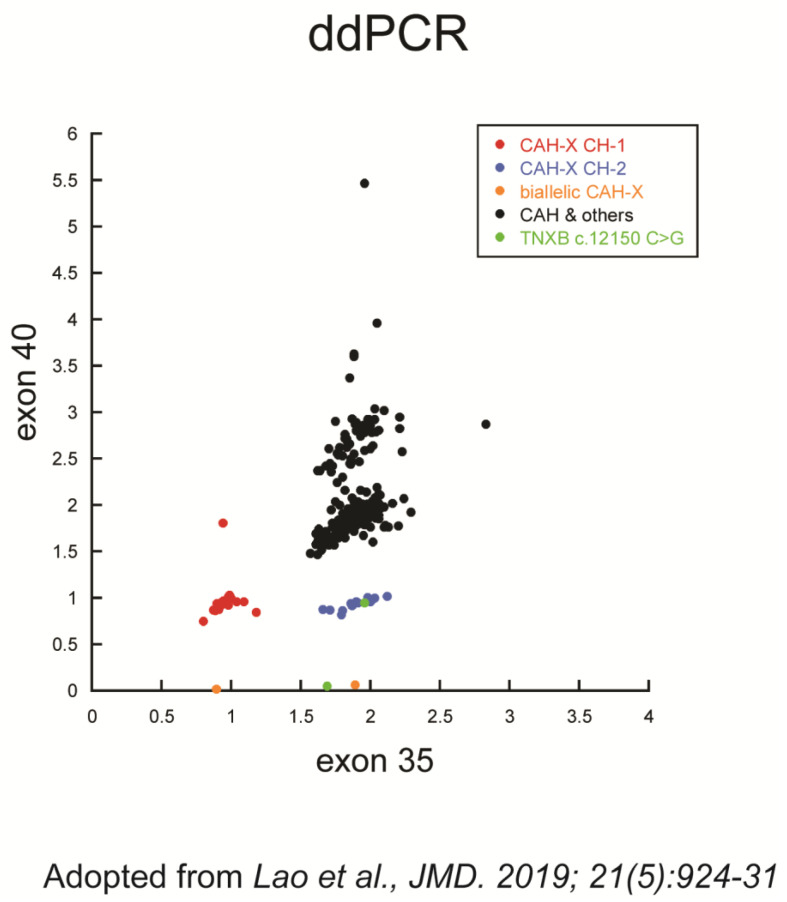
Identification of CAH-X chimeric type by droplet digital PCR (ddPCR) [10]. The copy numbers of *TNXB* exons 35 and 40 determined by ddPCR were used for the call of CAH-X chimeric genes.

**Table 1 genes-14-00265-t001:** *TNXB*-like *TNXA* variant alleles in subjects with excessive *TNXB* exon 40 copy number.

Subject#	CAH 21OHD or Other Disease Status	*CYP21A2* Genotype	*TNXB* X40 CNV (ddPCR)	Total *TNXA* CNV	*TNXA* Variant (X40-like)	In *cis CYP21A2* Allele
1	SW	In2G/In2G	5.47	3	3	In2G/In2G
2	SW	CAH-CH1/In2G	2.89	1	1	In2G
3	SW	R356W/In2G	2.79	3	1	N/D
4	SW	Q318X/CAH-CH5	2.62	1	1	Q318X
5	SW	In2G/CAH-CH5	2.66	1	1	In2G
6	SW	In2G/In2G	2.55	2	1	In2G
7	SW	In2G/Q318X	2.71	2	1	N/D
8	SW	In2G/CAH-CH1	2.87	1	1	In2G
9	SW	CAH-CH1/IVS8+1G>A	2.72	1	1	IVS8+1G>A
10	SV	CAH-CH1/I172N	2.64	1	1	I172N
11	SV	In2G/In2G	2.59	1	1	In2G
12	SV	CAH-CH1/del-I172N	2.81	2	1	del-I172N
13	SV	I172N/In2G	2.78	2	1	N/D
14	SV	In2G/E6cluster/I172N	2.5	2	1	N/D
15	SV	I172N/In2G	2.58	2	1	N/D
16	SV	I172N/Q318X	2.86	1	1	N/D
17	SV	I172N/Q318X	2.8	2	1	Q318X
12B	NC	del/del-I172N	2.74	2	1	del-I172N
2B	carrier	In2G/normal	2.61	2	1	In2G
4A	carrier	Q318X/normal	2.9	2	1	Q318X
6B	carrier	In2G/normal	2.76	2	1	In2G
8A	carrier	In2G/normal	2.92	2	1	In2G
11A	carrier	In2G/normal	3.37	2	2	In2G/normal
18A	carrier	In2G/normal	2.78	2	1	normal
19A	carrier	CAH-CH1/normal	2.83	1	1	normal
20A	carrier	CAH-CH1/normal	2.84	2	1	normal
21A	carrier	CAH-CH5/normal	2.55	2	1	normal
21C	carrier	P453S/normal	2.61	2	1	normal
22B	carrier	V281L/normal	2.56	3	1	normal
23A	carrier	CAH-CH3/normal	3.63	2	2	normal *
24A	carrier	In2G/normal	2.69	2	1	N/D
25A	carrier	I172N/normal	2.62	1	0	N/A
26B	carrier	CAH-CH5/normal	2.81	1	1	normal
27B	carrier	Q318X/normal	2.92	3	N/D ^†^	N/A
28B	carrier	V281L/normal	2.95	3	1	normal
29A	carrier	In2G/normal	2.77	2	1 ^‡^	normal
20C	normal	normal/normal	2.68	2	1	normal
30C	normal	normal/normal	2.54	2	1	normal
31A	normal	normal/normal	2.93	2	1	normal
32	XY DSD	normal/normal	3.6	2	2	normal/normal
33	Adrenal insufficiency	normal/normal	2.87	2	0	N/A ^§^
34	Aldosterone synthase deficiency	normal/normal	2.81	3	1	normal
35A	11βOHD carrier	normal/normal	2.92	2	1	normal
36	17OHD CAH	normal/normal	3.96	3	2	normal/normal ^¶^
37A	11βOHD carrier	normal/normal	2.86	2	1	normal

* Two *TNXA* variant alleles in cis with a normal *CYP21A2* within a tri-modular *RCCX* module. ^†^ Sequence analysis unavailable due to repeated PCR failures. ^‡^ A mono-allelic *TNXA* variant with only rs77471377(C>G). ^§^ This subject had two copies of wild-type *TNXA* allele and three copies measured in *TNXB* exon 35. The excessive copy numbers were likely due to a variant in the *HBB* gene whose copy number was used for normalization. ^¶^ This subject had two copies of *TNXA* variants and on copy of *TNXA* wild-type, not able to determine their haplotype as *TNXA(v)-TNXA(v)/TNXA(wt)* or *TNXA(v)-TNXA(wt)/TNXA(v)*. Abbreviations: 21OHD, 21-hydroxylase deficiency; SW: salt-wasting 21OHD; SV: simple virilizing 21OHD; NC: non-classic 21OHD; 11βOHD: 11-β_1_ hydroxylase deficiency; 17OHD: 17-hydroxylase deficiency; N/A: not applicable; N/D: not determined. The suffixes of A, B and C after the subject number denote mother, father and sibling of an affected patient, respectively. *CYP21A2* variants are shown in common names (ref. NM_000500.9): CAH-CHs: *CYP21A1P/CYP21A2* chimeric genes with known chimera type; del: 30-kb deletions with chimera unspecified; In2G: c.293-13A/C>G; I172N: c.518T>A; E6 cluster: c.710T>A, c.713T>A, c.719T>A; V281L: c.844G>T; Q318X: c.955C>T; R356W: c.1069C>T; IVS8+1G>A: c.1118+1G>A; P453S: c.1360C>T.

## Data Availability

The authors confirm that the data supporting the findings of this study are available with the article.

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
