# Peer review of "Pseudogene TNXA Variants May Interfere with the Genetic Testing of CAH-X"

_genes, 2023, doi:10.3390/genes14020265_

Round 1

Reviewer 1 Report

Dear the Authors

Reviewer has evaluated a paper by Lao et al., describing a risk of false results on CAH-X CH-2 caused by high-throughput screening such as ddPCR. The clinically-important point for variant detection in patients with CAH-X or classical-like EDS is thought to be whether the variant is present within TNXB or not. Generally, genetic testing for TNXB-related classical-like EDS is performed by Sanger sequencing using TNXB-specific primers or Sanger sequencing following long PCR using TNXB-specific primers for exons 3244, and Sanger sequencing or NGS for other regions (van Dijk et al., GeneReviews, 2022). Although it is scientifically interesting to consider the structural variations in the RCCX modules and the manuscript is well-written, several major and minor issues have been found as follows:

Major

1.      The authors described that “the hypermobile EDS connective tissue dysplasia due to TNXB defects appears to be autosomal dominant, as carrying one CAH-X allele has been associated with an EDS phenotype”. However, according to the 2017 International Classification (PMID: 28306229), the genetic basis of hypermobile EDS is still unknown (hypermobile EDS is not tenascin-X haplosufficiency syndrome), and TNXB gene is associated with a recessively-inherited classical-like EDS. Therefore, a subject with heterozygous CAH-X is a carrier for classical-like EDS.

2.      The authors stressed the efficacy of ddPCR as a high-throughput screening for CAH-X. However, a NGS-based screening (e.g., short read sequencing following specific amplification, long read sequencing) could be more high-throughput and accurate. What do the authors think about superiority of ddPCR to such NGS-based screening?

3.      It would be difficult to understand the relationship between copy number measurement and structural variation in some cases as follows:

Subject 25A: The authors described that the subject 25A had her TNXB exon 40 copy number measured as 2.62; she had one copy of TNXA which was determined as wild type that matches the reference genome in this study, thus the mechanism underlying the excessive 0.62 copy of TNXB exon 40 remains unclear. Would it be possible to interpret the rearrangement of the subject 25A as “RP2-C4B-CYP21A2-TNXB/TNXA”-“RP2-C4B-CYP21A2-TNXB” in one allele?

Subject 33: The authors described that the subject 33 had the TNXB exon 40 copy number measured as 2.87; two wild type copies of TNXA were present. This was the lone case of having TNXB exon 35 measured as 2.83 (Figure 2) without further evidence of structural variation in the loci, thus the excessive copy numbers were likely due to some variants in the HBB gene whose copy number was used to normalize the TNXB exons in the ddPCR assays. Would it be possible to interpret the rearrangement of the subject 33 as “RP1-C4A-CYP21A1P-TNXA”-“RP2-C4B-CYP21A2-TNXB/TNXA”-“RP2-C4B-CYP21A2-TNXB” in one allele?

Minor

Line 241, 242, 287:

“Multiplex ligation-dependent probe amplification (MPLA)” should be changed “MLPA”

Reviewer 2 Report

The present manuscript submitted by Lao et al. is an extended study of their previous work “High-Throughput Screening for CYP21A1P-TNXA/TNXB Chimeric Genes Responsible for Ehlers-Danlos Syndrome in Patients with Congenital Adrenal Hyperplasia” published in JMD. In the present manuscript, authors address the issues regarding the potential interference with CAH-X molecular genetic testing based on copy number assessment by digital PCR and multiplex ligation-dependent probe amplification, since this TNXA variant allele might mask a real copy number loss in TNXB exon 40. The authors analyzed the TNXA locus homologous to TNXB exon 40, using allele-specific PCR followed by Sanger sequencing. And also mentioned that the SNPs were determined in all heterozygous cases by either family genotyping or TA clone sequencing, reflecting the ddPCR methodology which detects SNPs in tandem. Overall, their finding suggests that the standard NGS-based methodologies are not suitable for the genetic testing of RCCX module genes within the homologous window, for  both CYP21A2 and TNXB exons 32-44. Therefore, authors finally  proposed that the pseudogenes CYP21A1P and TNXA, as well as their underestimated allele frequency, should be considered as an obstacle to overcome in addition to the commonly accepted pseudogene homolog and copy number variations in terms of designing a comprehensive CAH test platform based on NGS methodologies.

Minor Concern:

Authors should explain the “functional role of pseudogene TNXA variants in the manifestation of CAH-X and how their interference will impact the genetic testing of CAH-X” either in the discussion or introduction section. Overall, the information provided is valuable and well-presented in this manuscript. 

Round 2

Reviewer 1 Report

Dear the Authors

Reviewer has re-evaluated a paper by Lao et al., describing a risk of false results on CAH-X CH-2 caused by high-throughput screening such as ddPCR. Most of issues Reviewer pointed out in the previous review have been corrected appropriately. Reviewer would like to ask the authors to reconsider only one issue as follows:

2017 International Classification (PMID: 28306229) defined that hypermobile EDS is the subtype of EDS, the genetic basis of which is unknown and the diagnosis is solely based on clinical characteristics. The statement ”although the etiology for most hEDS cases remains unclear, a subset is due to defects in TNXB” would be misleading. 
